# Relief Effect of Carbon Dioxide Insufflation in Transnasal Endoscopy for Health Checks—A Prospective, Double-Blind, Case-Control Trial

**DOI:** 10.3390/jcm11051231

**Published:** 2022-02-24

**Authors:** Toshio Fujisawa, Hiroshi Fukuda, Naoto Sakamoto, Mariko Hojo, Ko Tomishima, Shigeto Ishii, Hirohide Yokokawa, Mizue Saita, Toshio Naito, Akihito Nagahara, Sumio Watanabe, Hiroyuki Isayama

**Affiliations:** 1Department of Gastroenterology, Graduate School of Medicine, Juntendo University, Tokyo 113-8421, Japan; t-fujisawa@juntendo.ac.jp (T.F.); sakamoto@juntendo.ac.jp (N.S.); mhojo@juntendo.ac.jp (M.H.); tomishim@juntendo.ac.jp (K.T.); sishii@juntendo.ac.jp (S.I.); nagahara@juntendo.ac.jp (A.N.); sumio@juntendo.ac.jp (S.W.); 2Department of General Medicine, Graduate School of Medicine, Juntendo University, Tokyo 113-8421, Japan; hiro@juntendo.ac.jp (H.F.); hyokoka@juntendo.ac.jp (H.Y.); msaita@juntendo.ac.jp (M.S.); naito@juntendo.ac.jp (T.N.)

**Keywords:** carbon dioxide, CO_2_ insufflation, abdominal pain, abdominal distention, transnasal endoscopy, health check

## Abstract

CO_2_ insufflation has proven effective in reducing patients’ pain after colonoscopies but has not been examined in esophagogastroduodenoscopies. Therefore, we examined the effect of CO_2_ insufflation in examinees who underwent transnasal endoscopies without sedation. This study is a single-center, prospective, double-blind, case-control trial conducted between March 2017 and August 2018. Subjects were assigned weekly to receive insufflation with either CO_2_ or air. The primary outcome was improvement of abdominal pain and distension at 2 h and 1-day postprocedure. In total, 336 and 338 examinees were assigned to the CO_2_ and air groups, respectively. Visual analog scale (VAS) scores for abdominal distension (15.4 vs. 25.5; *p* < 0.001) and distress from flatus (16.0 vs. 28.8; *p* < 0.001) at 2 h postprocedure were significantly reduced in the CO_2_ group. VAS scores for pain during the procedure (33.5 vs. 37.1; *p* = 0.059) and abdominal pain after the procedure (3.9 vs. 5.7; *p* = 0.052) also tended to be lower at 2 h postprocedure, but all parameters showed no significant difference at 1-day postprocedure. All procedures were safely completed through the planned program, and no apparent adverse events requiring treatment or follow-up occurred. In conclusion, CO_2_ insufflation may reduce postprocedural abdominal discomfort from transnasal esophagogastroduodenoscopies. (UMIN000028543).

## 1. Introduction

Carbon dioxide (CO_2_) is rapidly absorbed from the gastrointestinal tract and easily eliminated by respiration. It is absorbed approximately 160 times faster than nitrogen, which is the major gaseous ingredient of air [1]. CO_2_ insufflation has proven effective in reducing patients’ pain after endoscopic procedures. Its effect has been examined mainly in colonoscopies [2,3,4] and in small numbers, in balloon-assisted endoscopies [5] and endoscopic retrograde cholangiopancreatographies [6,7,8]. Regarding the upper gastrointestinal tract, the usefulness of CO_2_ in therapeutic endoscopies, such as endoscopic submucosal dissections (ESDs) [9,10], has been examined, but it has not been examined in diagnostic endoscopies. The principal source of pain from a colonoscopy is intestinal hyperextension by air insufflation, whereas in an esophagogastroduodenoscopy, pain is largely due to the vomiting reflex. Because the main causes of pain are different, it is difficult for the endoscopist to identify the pain relief effect of CO_2_ insufflation. This may be one reason that the effect of CO_2_ insufflation has not been studied in esophagogastroduodenoscopies. However, esophagogastroduodenoscopies also require considerable insufflation, similar to colonoscopies, so abdominal pain and distension may occur during and after the procedures in the same way.

Because screening esophagogastroduodenoscopies are performed on healthy examinees during health checks, it should be as painless as possible. Esophagogastroduodenoscopies during health checks are often performed by transnasal endoscopies to reduce patients’ discomfort [11], but few institutions perform endoscopies under sedation due to labor shortages and high costs [12]. In addition, patients who receive health check endoscopic examinations have no symptoms, so bias from previous symptoms is low.

Therefore, we examined the effect of CO_2_ insufflation on pain relief in examinees who underwent transnasal endoscopies without sedation.

## 2. Materials and Methods

This study is a prospective, case-control, double-blind trial; its protocol was approved by the institutional review board and the database is open to the public (UMIN000028543).

### 2.1. Inclusion and Exclusion Criteria

The inclusion criteria consist of subjects who were examinees who underwent esophagogastroduodenoscopies as part of health checks at our hospital from March 2017 to August 2018 and agreed to participate in the present study.

The exclusion criteria consist of the following: 1. examinees under 20 years of age and those unable to understand information about the purpose of the study; 2. examinees with severe chronic obstructive pulmonary disease (COPD) and known CO_2_ retention; 3. examinees who were unable to complete questionnaires at 2 h and 1 day after the procedure. If either questionnaire could not be collected, the subject was removed from the analysis.

### 2.2. Allocation and Blinding

After written agreement to the study procedures was obtained, the subjects were automatically assigned to receive insufflation with either CO_2_ (CO_2_ group) or air (air group). The selection of CO_2_ or air insufflation changed weekly. This method was chosen to eliminate any bias between the groups attributable to the endoscopist because the attending endoscopist was determined by the day of the week. The schedule was established before the study began and was not changed during the study. A weekly gas exchange was chosen to avoid bias from the endoscopist among groups and to avoid unblinding by changing the gas between procedures. All persons directly involved in the endoscopies, including the examinees, the endoscopists, and the nurses, were blinded with a blindfold on the inflation machine regarding which gas was used.

### 2.3. Endoscopic Procedure

The transnasal endoscopies were performed with video scopes (EG-L580NW; Fujifilm corporation, Tokyo, Japan) with only local anesthetics to the nostrils. Nostril patency was tested using a pretreatment transnasal catheter of comparable scope diameter (N10/18F-W 6.0 mm-6 cm; TOP Corporation, Tokyo, Japan) lubricated with 2% lidocaine gel. If the nostrils were narrow and the endoscope could not be inserted, a nasal endoscope was inserted orally after 50–100 mg of Xylocaine spray (2%) was added to the oral cavity as a local anesthetic. All procedures were performed without any sedative agents because the endoscopy facility did not use sedatives in transnasal endoscopies for health checks. In the endoscopic examination, the pharynx, esophagus, stomach, duodenal bulb, and second portion were observed, and a biopsy was performed when histopathological examination was necessary. Even if a lesion requiring treatment was found, therapeutic endoscopies were performed on another day. In total, 26 endoscopists and 19 nurses oversaw the endoscopic examinations.

### 2.4. Gas Delivery

CO_2_ was delivered using a CO_2_ regulator (GW-1; Fujifilm). The gas flow rate was set at 1.8 L/min for both CO_2_ and air insufflation. To prevent unblinding, the regulator was placed behind the endoscopy rack and hidden from the endoscopist’s view. The gas to be delivered was set by the coordinator before all procedures of the day began.

### 2.5. Patient Questionnaire

Examinees completed questionnaires at 2 h and at 1-day post-procedure. The 2 h questionnaire was completed at the hospital before leaving for home, and the 1-day questionnaire was filled out at home and returned by mail. Cases in which either questionnaire could not be retrieved were excluded from the analysis. A visual analogue scale (VAS) was used to rate aspects of the procedure from 0 to 100, with 0 being the lowest and 100 being the highest rating for each factor. Patients rated four factors of their procedural experiences on each questionnaire. The questionnaire at 2 h post-procedure asked about pain during the endoscopy, abdominal distension, abdominal pain, and distress from flatus. The questionnaire at 1-day post-procedure asked about the patient’s preferences during a future endoscopy instead of pain during this endoscopy. Both questionnaires had space for free-form comments about other discomfort.

### 2.6. Endoscopist/Nurse Questionnaire

The endoscopists and the attending nurses independently completed questionnaires to objectively assess patient discomfort using a score from 0 to 10, with 0 indicating no discomfort and 10 indicating maximum discomfort. The endoscopist also evaluated the overall procedural aspects of scope handling on a scale from 0 to 10, with 0 indicating the easiest and 10 the most difficult possible procedure. This evaluation was performed immediately after the procedure. In addition to the information provided in the questionnaires, participants’ age, sex, length of examination, history of endoscopies, presence of biopsy, and rate of change from a nasal to an oral route were also included in the study.

### 2.7. Outcomes and Statistical Analysis

The primary outcome was set as improvement of abdominal pain and distension at 2 h and 1-day post-procedure. These data were collected by questionnaire from the patients after the procedure. The secondary outcomes were set as painfulness of endoscopies, procedure times, and rate of adverse events. The results of the questionnaires from the endoscopists and nurses were also analyzed.

We were planning a study of a continuous response variable in control and experimental subjects with one control for each experimental subject. In a previous study [9,10], the responses within each subject group were normally distributed with a standard deviation of 40. If the true difference between the experimental and control means were 10, we would need to study 337 experimental subjects and 337 control subjects to be able to reject the null hypothesis that the population means of the experimental and control groups are equal with a probability (power) of 0.9. The type I error probability associated with this test of the null hypothesis is 0.05. Analyses were performed on a per-protocol basis for patients who underwent the procedure. Characteristics of the study groups were compared with a t-test or Mann–Whitney U test for continuous variables and a chi-squared test (or Fisher’s exact test, as appropriate) for categorical variables. A two-sided *p*-value < 0.05 was considered statistically significant for all tests.

## 3. Results

### 3.1. Subject Allocation

From March 2017 to August 2018, 1794 examinees underwent transnasal endoscopies during health checks at our hospital. Twenty-two examinees were excluded from the study due to a history of COPD. The study period was 72 weeks, with 36 weeks each allocated to the CO_2_ and air groups. The average number of participants per week was 9.3. Finally, 674 examinees agreed to participate in the present study; 336 and 338 were allocated to the CO_2_ group and air group, respectively. The flow chart of subject allocation is shown in Figure 1.

### 3.2. Participant Characteristics

Participant characteristics and features of endoscopic procedures in the groups receiving CO_2_ and air insufflation are shown in Table 1. There were no significant differences between the groups in any parameters, including sex, age, experience of transnasal endoscopy, length of procedure, rate of biopsy, and rate of change to oral endoscopy.

### 3.3. Results of Examinee Questionnaires

The results of examinee questionnaires are summarized in Table 2 and Figure 2A–C. The VAS scores for abdominal distension (15.4 vs. 25.5; *p* < 0.001) and distress from flatus (16.0 vs. 28.8; *p* < 0.001) at 2 h post-procedure were significantly reduced in the CO_2_ group compared to the air group, respectively. With regard to pain during the procedure and abdominal pain afterward, the VAS scores of the CO_2_ group tended to be lower than those of the air group at 2 h post-procedure, but the difference was not significant (33.5 vs. 37.1, respectively; *p* = 0.059 for pain during the procedure; 3.9 vs. 5.7, respectively; *p* = 0.052 for postprocedure abdominal pain). At 1-day post-procedure, there were no differences between the CO_2_ and air groups, respectively, in any parameter including abdominal distension (8.1 vs. 7.0; *p* = 0.383), distress from flatus (11.4 vs. 12.5; *p* = 0.490), abdominal pain (4.1 vs. 3.0; *p* = 0.166), and preference in a future endoscopy (19.1 vs. 18.8; *p* = 0.844). For abdominal distension, distress from flatus, and abdominal pain, the change in VAS score over time from 2 h to the next day is shown in Figure 2A–C.

Some examinees complained of nasal symptoms such as nasal pain and rhinorrhea in the free comments at both 2 h and 1-day post-procedure. The rate of nasal symptoms was compared between the groups (no data shown). However, there was no significant difference between CO_2_ and air insufflation at both time points (9.4% vs. 9.8%; *p* = 0.869 at 2 h post-procedure and 4.9% vs. 3.5%; *p* = 0.355 at 1-day postprocedure, respectively).

### 3.4. Results of Endoscopist and Nurse Questionnaires

The results of the endoscopist and nurse questionnaires are summarized in Table 3. Ratings of overall scope handling by the endoscopist showed no significant difference between the CO_2_ group and the air group (1.75 vs. 1.79; *p* = 0.709). The endoscopist and attending nurse evaluations of patient discomfort also showed no significant difference between the two groups (1.93 vs. 2.00; *p* = 0.471 by endoscopist, 1.29 vs. 1.32; *p* = 0.794 by nurse).

### 3.5. Adverse Events

All procedures were safely completed according to the planned program, and there were no apparent adverse events requiring treatment or follow-up.

## 4. Discussion

### 4.1. Evaluation of the CO_2_ Effect on Reducing Abdominal Discomfort

This prospective, double-blind, case-controlled study revealed that CO_2_ insufflation during transnasal esophagogastroduodenoscopies reduced postprocedural abdominal discomfort, including abdominal distension and distress from flatus, compared to air insufflation. Initially, a randomized-control trial was planned, but the present study involved healthy people undergoing health checks, and a large number of examinees received endoscopies on the same day. Therefore, we were unable to take the time to randomize the subjects and double-blind them, and finally abandoned randomization. The VAS scores at 2 h post-procedure decreased by about 50 to 60% in the CO_2_ insufflation group, although VAS scores at 1-day post-procedure were not significantly different. Many previous studies and meta-analyses have also reported that CO_2_ insufflation reduces postprocedural abdominal pain compared to air insufflation. However, the targeted procedures were limited to colonoscopies and therapeutic endoscopies, such as double-balloon endoscopies, endoscopic retrograde cholangiopancreatoscopies, and ESDs. This is the first study to demonstrate the usefulness of CO_2_ insufflation in a brief diagnostic endoscopy, the transnasal esophagogastroduodenoscopy. In the present study, pain during the procedure, abdominal distension, distress from flatus, and abdominal pain were used as indices to evaluate abdominal discomfort, almost the same as the parameters used in previous studies. Pain during the procedure tended to be lower with CO_2_ insufflation, but the difference was not statistically significant. The difference between CO_2_ and air seemed to be small because nasal pain and the vomiting reflex had greater effects than gas insufflation on patient discomfort from transnasal esophagogastroduodenoscopies. The examinees’ discomfort was significantly different between the CO2 and air groups as assessed by the subjects themselves using the VAS scores but not by the nurses. These results indicate that the subjective evaluation of the examinees their selves is important for the discomfort from endoscopies because the objective evaluation by the surrounding medical personnel is not accurate.

### 4.2. Duration of the CO_2_ Effect on Reducing Abdominal Discomfort

In the present study, CO_2_ was observed to reduce abdominal discomfort at 2 h after the procedure but not on the next day. Rogers et al. analyzed the duration of the CO_2_ effect on postprocedural pain in their meta-analysis [13]. This meta-analysis, which included 23 studies, revealed that patients receiving CO_2_ insufflation consistently had less pain at 2 h postprocedure, and this effect persisted at 6 h postprocedure. However, there was no difference in abdominal pain between the CO_2_ and air groups at 24 h postprocedure. Dollen et al. also reported that the CO_2_ effect persisted to 6 h after the procedure in their meta-analysis [14]. The results of these meta-analyses match those of the present study. Therefore, the advantage of CO_2_ insufflation in esophagogastroduodenoscopies can be expected to last for about 6 h after the procedure. The COVID-19 pandemic has become a worldwide concern in recent years, as has endoscopy-related transmission. CO_2_ insufflation may reduce the length of the patient’s hospital stay by reducing abdominal discomfort on the day of the test, and the reduced length of stay might help limit COVID-19 transmission.

### 4.3. CO_2_ Effect on Procedure Time

Because the absorption of CO_2_ from the gastrointestinal tract is much faster than that of air, there was concern that the amount and frequency of CO_2_ insufflation required for the procedure would be greater, resulting in a longer procedure time. However, the procedure time did not differ between CO_2_ insufflation and air insufflation (8.98 min. vs. 8.95 min, respectively; *p* = 0.893). Previous randomized controlled trials (RCTs) [15,16,17] and meta-analyses [13] showed similar results in colonoscopies. Therefore, CO_2_ insufflation seems to have little effect on the procedure time of transnasal esophagogastroduodenoscopies.

### 4.4. Safety of CO_2_ Insufflation during Transnasal Esophagogastroduodenoscopies

Carbon dioxide is easily eliminated by respiration after absorption from the gastrointestinal tract. Since the respiratory system is the main route of CO_2_ excretion, most prospective studies, including the present study, excluded examinees with severe respiratory compromise or COPD with known CO_2_ retention. Although severe respiratory compromise has been excluded, there have been no reports of adverse events with CO_2_ insufflation in previous prospective studies. In the present study of 674 participants, no apparent adverse events were observed except for nasal hemorrhage that did not require treatment. Rogers et al. examined CO_2_ concentration in the arterial blood before and after colonoscopic polypectomy with CO_2_ insufflation and found that the increase in CO_2_ partial pressure was limited (37.3–40.6 mmHg), and the pH value did not change (from 7.46 to 7.45) [18]. Bretthauer et al. evaluated end-tidal CO_2_ under CO_2_ insufflation in colonoscopies and revealed that end-tidal CO_2_ decreased after the procedure with both CO_2_ insufflation and air insufflation [19]. The results of these studies suggest that CO_2_ insufflation during health check esophagogastroduodenoscopies is sufficiently safe.

### 4.5. Cost of CO_2_ Insufflation for Esophagogastroduodenoscopies during Health Checks

Many RCTs and meta-analyses have examined the utility of CO_2_ insufflation in diagnostic and therapeutic endoscopies, and its efficacy is now well established. However, in a 2009 survey of 580 European endoscopists, only 47% had heard of CO_2_ insufflation. Moreover, among them, 87% were aware of the convincing evidence showing that CO_2_ was superior to air, but only 4% were using CO_2_ insufflation [20]. Cost is one of the reasons that CO_2_ insufflation is not widely used [21]. Air insufflation does not cost money, but CO_2_ insufflation does. No study addressed the additional cost of CO_2_ use for endoscopic insufflation, and to date, no cost-effectiveness study has been performed. In practice, the cost of installing a CO_2_ insufflator is about $4000 to $8000, but once it is installed, the running cost is negligible [22]. In this study, procedure time was about 9 min in the CO_2_ group, and the amount of CO_2_ used in one procedure was 16.2 L, even if CO_2_ was delivered throughout the procedure. Given that liquid CO_2_ can be purchased for about $2 per kilogram, the cost of CO_2_ per procedure can be calculated as about $0.06. There was no significant difference in preference in a future endoscopy between CO_2_ and air insufflation, so the likely impact of increasing the use of CO_2_ seems to be minimal. In addition, in areas where medical resources are limited, the installation of CO_2_ regulators and the stable supply of CO_2_ could be difficult at present, and therefore, the use of CO_2_ in routine clinical practice is limited in those areas. However, there is no reason to be reluctant to use CO_2_, especially during health checks, because it can reduce abdominal discomfort on the day of the endoscopy at a low cost.

### 4.6. Limitations

First, it was conducted at a single center. Second, the subjects were not assigned randomly. The selection between CO_2_ and air insufflation was changed weekly, and the schedule was already established before the study began. Although this technique is not randomization, bias is considered to be low. Third, the rate of obtaining consent was low, 38%, because the study was conducted with examinees having regular health checks.

While recognizing these limitations, we recommend CO_2_ insufflation during transnasal esophagogastroduodenoscopies to reduce abdominal discomfort after the procedures.

## 5. Conclusions

CO_2_ insufflation may reduce postprocedural abdominal discomfort in transnasal esophagogastroduodenoscopy.

## Figures and Tables

**Figure 1 jcm-11-01231-f001:**
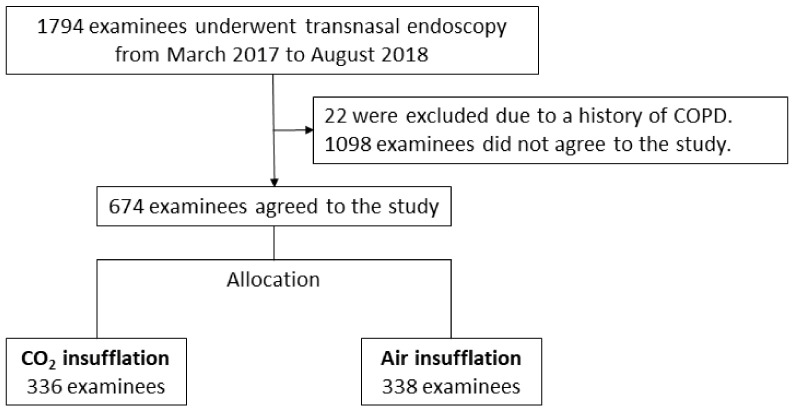
Flow chart of examinee allocation. Of the total 1794 examinees, 674 were finally included in the analysis.

**Figure 2 jcm-11-01231-f002:**
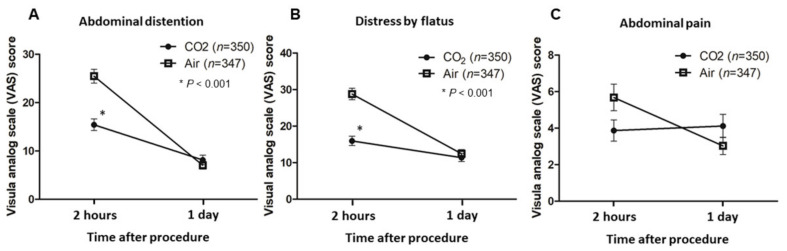
The change over time in the visual analog scale (VAS) score from 2 h to 1 day after the procedure. (**A**) VAS scores for abdominal distension. The VAS score for CO_2_ insufflation was significantly lower than that for air insufflation (15.4 vs. 25.5; *p* < 0.001) at 2 h, but not at 1 day after the procedure. (**B**) VAS scores for distress due to flatus. The VAS score for CO_2_ insufflation was significantly lower than that for air insufflation at 2 h after the procedure (16.0 vs. 28.8; *p* < 0.001). (**C**) VAS scores for abdominal pain. There was no significant difference in abdominal pain between the two groups.

**Table 1 jcm-11-01231-t001:** Patient characteristics and features of the endoscopic procedures in the groups receiving carbon dioxide (CO_2_) and air insufflation.

	CO_2_ Group (*n* = 336)	Air Group (*n* = 338)	*p* Value
Sex, M/F	212/124	205/133	0.526
Age (year) *	61.2 ± 11.5	60.1 ± 11.8	0.206
Transnasal endoscopy experienced	167 (49.7%)	178 (52.6%)	0.488
Procedure time (min) *	8.98 ± 2.87	8.95 ± 2.93	0.895
Biopsy performed	33 (9.8%)	43 (12.7%)	0.273
Change to oral	21 (6.2%)	23 (6.8%)	0.876

* expressed as mean ± standard deviation.

**Table 2 jcm-11-01231-t002:** Results of examinees’ questionnaires.

Timing	Abdominal Distention			Distress by Flatus			Abdominal Pain		
	CO_2_	Air	*p* Value	CO_2_	Air	*p* Value	CO_2_	Air	*p* Value
2 h	15.4 ± 1.1	25.5 ± 1.4	<0.001 *	16.0 ± 1.3	28.8 ± 1.6	<0.001 *	3.9 ± 0.6	5.7 ± 0.7	0.052
1 day	8.1 ± 1.0	7.0 ± 0.8	0.383	11.4 ± 1.1	12.5 ± 1.1	0.490	4.1 ± 0.6	3.0 ± 0.5	0.166
**Timing**	**Painfulness during Procedure**			**Preference in a Future Endoscopy**					
	**CO_2_**	**Air**	***p* Value**	**CO_2_**	**Air**	***p* Value**			
2 h	33.5 ± 1.4	37.1 ± 1.3	0.059	-	-	-			
1 day	-	-	-	19.1 ± 1.4	18.8 ± 1.3	0.844			

Data was expressed as mean ± standard error. * shows significant difference between CO_2_ and air insufflation at same timing.

**Table 3 jcm-11-01231-t003:** Results of endoscopist and nurse questionnaires.

Evaluator	Parameter	CO_2_ (*n* = 336)	Air (*n* = 338)	*p* Value
Endoscopist	Overall scope handling	1.75 ± 0.08	1.79 ± 0.08	0.709
	Examinee’s discomfort	1.93 ± 0.08	2.00 ± 0.08	0.471
Nurse	Examinee’s discomfort	1.29 ± 0.06	1.32 ± 0.06	0.794

Data was expressed as mean ± standard error. Each parameter was evaluated on a scale of 10.

## Data Availability

The data sets used and/or analyzed during the current study are available from the corresponding authors on reasonable request. However, the data sets include personal information. Therefore, limited information without personal information is available.

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
