# Peer review of "Relief Effect of Carbon Dioxide Insufflation in Transnasal Endoscopy for Health Checks—A Prospective, Double-Blind, Case-Control Trial"

_jcm, 2022, doi:10.3390/jcm11051231_

Round 1
Reviewer 1 Report
First of all I want to congratulate the authors with their hard work.
Overall I find the paper well-written with a fine english language.
However I have a few issues that need to be adressed.
- Table 2 is "squezzed" so the content looks a bit odd with words and numbers divided incorrect.
- Due to the extra costs for CO2 as compared to air, countries with limited health resources may find it impossible to implement the use of CO2 in clinical practice. I think you need to add this to discussion section.
- Please add a few lines to the Discussion section to clarify why you did not conduct a RCT
Author Response
Response to comments from Reviewer 1
First of all, I want to congratulate the authors with their hard work. Overall I find the paper well-written with a fine English language. However, I have a few issues that need to be addressed.
- Table 2 is "squeezed" so the content looks a bit odd with words and numbers divided incorrect.
Response: We would like to express our deep gratitude to the reviewer for reading our paper seriously and appreciating our study. We also appreciate the valuable comments to improve the paper. We have modified Table 2 following the reviewer’s comment.
- Due to the extra costs for CO2 as compared to air, countries with limited health resources may find it impossible to implement the use of CO2 in clinical practice. I think you need to add this to discussion section.
Response: Thank you to the reviewer for pointing out a very important point. The following sentence was added to the discussion part.
“In addition, in areas where medical resources are limited, the installation of CO2 regulator and the stable supply of CO2 could be difficult at present, and therefore the use of CO2 in routine clinical practice is limited in those areas.”
- Please add a few lines to the Discussion section to clarify why you did not conduct a RCT
Response: The present study involved healthy people undergoing medical checkups, and we had to perform endoscopy on a large number of people on the same day. Because of this, we were unable to take the time to randomize the subjects and double-blind them, so we abandoned randomization. The following sentence was added to the discussion part.
“Initially, a randomized-control trial was planned, but the present study involved healthy people undergoing health check, and a large number of examinees received endoscopy on the same day. Therefore, we were unable to take the time to randomize the subjects and double-blind them, and finally abandoned randomization.”

Reviewer 2 Report
It was a pleasure to review the manuscript by Fujisawa et al. The authors conducted what appears to be a single-centre, double-blind, randomised controlled trial (674 patients) to assess the impact of CO2 versus standard air insufflation on comfort outcomes in patients undergoing unsedated transnasal upper GI endoscopy. The authors used visual analogue scales and found lower rates of abdominal discomfort and distress with CO2 within the early postprocedural period (within 2 hours), but not beyond this. This is an important study that may have practice implications on patient undergoing EGD, and such data are limited in the literature.
In my view, the paper is very well-written and the statistical analyses and methods appear robust, with no baseline differences between the two groups. The results are also credible and confirm the suspicions of many endoscopists, i.e. that CO2 insufflation, especially just before colonoscopy, may be better tolerated by patients. It should be emphasised that procedures times in Japan (i.e. mean of 9 minutes) may be longer than Western practice but is a standard that we are aspiring towards. The discussion is also beautifully structured and well-articulated.
I wish to make the following suggestions and comments:
Major
- Title: If this is a randomised controlled trial, this should be stated in the title as per CONSORT guidelines. References to “case-control trial” should be removed from the manuscript. This would carry higher level of evidence compared to a case-control study. I would also revise the title to give this paper more impact, i.e. “Effect of Carbon dioxide insufflation versus air insufflation on patient comfort in transnasal upper gastrointestinal endoscopy – a double blind, randomised controlled trial.”
- A “Conclusion” section would be helpful in the abstract and manuscript text.
- Results: Table 2 refers to the primary outcome. This is very difficult to read and interpret. Please consider converting this into a Figure, or adding graphical data above the table to give the study more visual impact.
- The discomfort visual analog score was scored from 0-100. Is there a threshold where discomfort levels would be considered as high (e.g. >50)? This may enable comparisons of the number of patients with significant discomfort (as a percentage) in CO2 vs air groups.
- Discussion: This should bring home the practice implications of the findings, including the fact that this study was performed in the pre-COVID era. With COVID infection control precautions and efforts to minimise endoscopy-related hospital stay, this study becomes even more relevant – this should be included under discussion.
Minor
- The methods section stated that evaluation of patient discomfort by the nurse assistant was performed immediately post procedure, but this showed no difference in discomfort levels between patients receiving CO2 and air, which goes against your primary outcome data. How do the authors explain the discrepancy? Perhaps this adds to our current knowledge that the objective assessment of pain (by investigators) is inaccurate and this is better characterised by patient questionnaires.
- Did any patients require sedation? If excluded, this should be made clear, as it introduces bias. If this is 0% in both groups, this should also be stated.
- I would consider a questionnaire completed at 2 hours (vs immediately postprocedure) to be prone to recall bias. Is there a reason specifically for 2 hours when nurse-rated discomfort levels were done post-procedurally?
- What are the authors’ thought on transnasal gastroscopy and whether this is an aerosol generating procedure? This could be included under discussion.
Thank you for this fine piece of work. I wish you all the very best with your submission.
Author Response
Response to the comments from Reviewer 2
It was a pleasure to review the manuscript by Fujisawa et al. The authors conducted what appears to be a single-centre, double-blind, randomised controlled trial (674 patients) to assess the impact of CO2 versus standard air insufflation on comfort outcomes in patients undergoing unsedated transnasal upper GI endoscopy. The authors used visual analogue scales and found lower rates of abdominal discomfort and distress with CO2 within the early postprocedural period (within 2 hours), but not beyond this. This is an important study that may have practice implications on patient undergoing EGD, and such data are limited in the literature.
In my view, the paper is very well-written and the statistical analyses and methods appear robust, with no baseline differences between the two groups. The results are also credible and confirm the suspicions of many endoscopists, i.e. that CO2 insufflation, especially just before colonoscopy, may be better tolerated by patients. It should be emphasised that procedures times in Japan (i.e. mean of 9 minutes) may be longer than Western practice but is a standard that we are aspiring towards. The discussion is also beautifully structured and well-articulated.
I wish to make the following suggestions and comments:
Major
- Title: If this is a randomised controlled trial, this should be stated in the title as per CONSORT guidelines. References to “case-control trial” should be removed from the manuscript. This would carry higher level of evidence compared to a case-control study. I would also revise the title to give this paper more impact, i.e. “Effect of Carbon dioxide insufflation versus air insufflation on patient comfort in transnasal upper gastrointestinal endoscopy – a double blind, randomised controlled trial.”
Response: I would like to thank the reviewer very much for reading our manuscript carefully and for your valuable comments. This study is a prospective, two-arm trial, but unfortunately it is not a randomized-control trial. This study involved healthy people undergoing medical checkups, and we had to perform endoscopy on a large number of people on the same day. Because of this, we were unable to take the time to randomize the subjects and double-blind them, so we abandoned randomization. The following sentence was added to the discussion part.
“Initially, a randomized-control trial was planned, but the present study involved healthy people undergoing health check, and a large number of examinees received endoscopy on the same day. Therefore, we were unable to take the time to randomize the subjects and double-blind them, and finally abandoned randomization.”
- A “Conclusion” section would be helpful in the abstract and manuscript text.
Response: Thank you for your important comment. We added the conclusion in both the abstract and the main text.
- Results: Table 2 refers to the primary outcome. This is very difficult to read and interpret. Please consider converting this into a Figure, or adding graphical data above the table to give the study more visual impact.
Response: As the reviewer pointed out, Table 2 is the most important result of the present study. Figure 2A-C is a graphical representation of three of the most important factors in this table: abdominal distance, stress from flat, and abdominal pain. In order to make the content easier for the reader to understand, we have added the descriptions in Figure 2A-C to the main text and adjusted the positions of Figure 2A-C and Table 2.
- The discomfort visual analog score was scored from 0-100. Is there a threshold where discomfort levels would be considered as high (e.g. >50)? This may enable comparisons of the number of patients with significant discomfort (as a percentage) in CO2 vs air groups.
Response: It is an important point and very instructive. The Visual analog scale is completely subjective to the examinee, and it is very difficult to draw the line between uncomfortable and not uncomfortable. In the next study, we will add questions that can be answered with two choices: uncomfortable or not uncomfortable.
- Discussion: This should bring home the practice implications of the findings, including the fact that this study was performed in the pre-COVID era. With COVID infection control precautions and efforts to minimise endoscopy-related hospital stay, this study becomes even more relevant – this should be included under discussion.
Response: As the reviewer pointed out, the condition after the endoscopy is improved by using CO2 insufflation, and it may play a role in preventing the spread of COVID-19. Therefore, the following description was added to the discussion.
“The COVID-19 pandemic has become a worldwide concern in recent years, as has endoscopy-related transmission. CO2 insufflation may reduce the length of the patient's hospital stay by reducing abdominal discomfort on the day of the test, and the reduced length of stay might help limit COVID-19 transmission.”
Minor
- The methods section stated that evaluation of patient discomfort by the nurse assistant was performed immediately post procedure, but this showed no difference in discomfort levels between patients receiving CO2 and air, which goes against your primary outcome data. How do the authors explain the discrepancy? Perhaps this adds to our current knowledge that the objective assessment of pain (by investigators) is inaccurate and this is better characterised by patient questionnaires.
Response: I appreciate your excellent point. We considered a similar consideration from these results. Therefore, the following sentence was added to the discussion part.
“Examinee’s discomfort was significantly different between the CO2 and air groups as assessed by the subjects themselves using the VAS scores, but not by the nurses. these results indicates that the subjective evaluation of the examinees himself is important for the discomfort by endoscopy because the objective evaluation by the surrounding medical personnel is not accurate.”
- Did any patients require sedation? If excluded, this should be made clear, as it introduces bias. If this is 0% in both groups, this should also be stated.
Response: In our facility, we do not sedate the examinees who undergo transnasal endoscopy for medical checkup. Therefore, all subjects in the present study were non-sedated. The following sentence was added to the method part for clarity:
“, because the endoscopy facility did not use sedatives in transnasal endoscopy for health check.”
- I would consider a questionnaire completed at 2 hours (vs immediately postprocedure) to be prone to recall bias. Is there a reason specifically for 2 hours when nurse-rated discomfort levels were done post-procedurally?
Response: Since the examinees are undergoing a health check and performing other examinations than endoscopy, there is a time difference between the time after endoscopy and the time when the VAS paper is collected. The VAS paper was collected when the result was explained after the endoscopic examination.
- What are the authors’ thought on transnasal gastroscopy and whether this is an aerosol generating procedure? This could be included under discussion.
Response: We think that both transnasal and oral endoscopy are likely to produce aerosols. Therefore, after the outbreak of COVID-19 infection, all endoscopic examinations are performed safely using the aerosols-diffusion-prevention device developed by us. I would appreciate it if you could read our video report on our endoscopy.
“A novel endoscopic shield: a barrier device to minimize virus transmission during endoscopy.” Endoscopy 2021 Dec 15. doi: 10.1055/a-1695-3080
Thank you for this fine piece of work. I wish you all the very best with your submission.

Round 2
Reviewer 2 Report
Thank you for your responses and for your revisions.